# Brimonidine Eye Drops within the Reach of Children: A Possible Foe

**DOI:** 10.3390/children11030317

**Published:** 2024-03-07

**Authors:** Daniela Trotta, Mirco Zucchelli, Carmela Salladini, Patrizia Ballerini, Claudia Rossi, Maurizio Aricò

**Affiliations:** 1Department of Pediatrics, Santo Spirito Hospital, Azienda Sanitaria Pescara, 65121 Pescara, Italy; daniela.trotta@asl.pe.it (D.T.); carmela.salladini@asl.pe.it (C.S.); 2Center for Advanced Studies and Technology (CAST), “Gabriele d’Annunzio” University of Chieti-Pescara, 66100 Chieti, Italy; m.zucchelli@unich.it (M.Z.); patrizia.ballerini@unich.it (P.B.); claudia.rossi@unich.it (C.R.); 3Department of Innovative Technologies in Medicine and Dentistry, “Gabriele d’Annunzio” University of Chieti-Pescara, 66100 Chieti, Italy

**Keywords:** brimonidine, eye drop poisoning, tandem mass spectrometry

## Abstract

Brimonidine, a selective alpha-2 adrenergic agonist used for the treatment of open-angle glaucoma, has been shown to cause neurological side effects such as unresponsiveness, lethargy, hypoventilation, and stupor, mimicking opioid toxicity. We report one case of transient encephalopathy in a toddler, in whom accidental brimonidine toxicity was suspected and then confirmed by a toxicology study. The healthy 8-month-old girl was taken to the pediatric ER since she was drowsy and hypotonic with miosis. The computed tomography scan of her brain and toxicological workup of her blood and urine were negative. Starting from the fourth hour, the child progressively improved, and by the sixth hour, she recovered to a normal state of consciousness. A survey of available drugs within the child’s reach showed the presence of brimonidine. Thus, ultra-performance liquid chromatography–tandem mass spectrometry (UPLC-MS/MS) was applied to quantify the brimonidine in urine and plasma samples, showing levels of 8.40 ng/mL and 0.79 ng/mL, respectively. To our knowledge, this is the first report to determine brimonidine levels in urine and plasma using UPLC-MS/MS. Insufficient knowledge on the part of family members about the potential hazards of an apparently innocuous, topical medication such as eye drops may put children at a greater risk of poisoning. Necessary warnings should be given to parents with greater care when prescribing this medication.

## 1. Introduction

Brimonidine is a selective alpha-2 adrenergic agonist, with a 1780-fold selectivity for α2- vs. α1-adrenergic receptors [1], and is increasingly used for the treatment of open-angle glaucoma (OAG) in children and adults [2]. After cataracts, glaucoma is the second leading cause of blindness in the world [3]. Open-angle glaucoma is the most common type of glaucoma among White and Black populations [3,4]. The estimated number of people with open-angle glaucoma in 2015 was 57.5 million, and this number was projected to increase to 65.5 million by 2020 [5]. The incidence of open-angle glaucoma increases with age, particularly in White and Black patients [6,7,8]. The prevalence of open-angle glaucoma is <1 percent in individuals under 55 years of age, and it approaches 2 percent at age 65 and reaches approximately 4 percent at age 80 [9].

Brimonidine is reported to be up to 12-fold more alpha-2-selective than clonidine and up to 32-fold more alpha-2-selective than apraclonidine (p-aminoclonidine) [10].

Brimonidine lowers intraocular pressure through a dual mechanism. It inhibits adenylyl cyclase with a decrease in cyclic adenosine monophosphate (AMP) levels and noradrenaline release through alpha-2 receptor activation, thus reducing aqueous humor production [11] and stimulating the outflow of aqueous humor via the uveoscleral pathway [12]. This latter mechanism is responsible for the drug effect following chronic treatment [13]. Although more polar and less lipophilic than clonidine (Figure 1), brimonidine is known to cross the blood–brain barrier [14].

Indeed, in addition to local complications such as dermatitis, brimonidine has been shown to cause systemic toxicity, including neurological side effects [15]. The potential central nervous system (CNS) side effects of α2-adrenergic stimulation in young children are unresponsiveness, lethargy, hypoventilation, and stupor. CNS depression may mimic opioid toxicity [16,17,18]. Berlin and coworkers [19] reported a case of a 1-month-old infant affected by Peters anomaly, characterized by corneal opacification associated with glaucoma and who presented recurrent episodes of unresponsiveness, hypotension, hypotonia, hypothermia, and bradycardia. The infant was under treatment with the following ophthalmic drops: Trusopt (dorzolamide), Betoptic (be-taxolol), and Alphagan (brimonidine). During each episode, following naloxone administration, the patient dramatically but only transiently improved. The episodes disappeared only after brimonidine discontinuation [19].

There are only limited data on the safety of brimonidine for children. The initial clinical trials mainly involved patients aged 2–7 years with a diagnosis of glaucoma, and no trials enrolled patients aged less than 1 year [10]. However, as it does not cause reduced pulmonary function and heart rate (HR), unlike β-adrenergic blocking agents, it is licensed for use in children over 5 years old [20]. Yet, a child’s lower weight could make the patient more susceptible to complications; thus, the pediatric age group is at greater risk [21,22]. In Italy, brimonidine administration to children aged 2 to 12 years is not recommended.

We report one case of transient encephalopathy in a toddler, in whom accidental brimonidine toxicity was suspected and then confirmed by a toxicology study.

## 2. Case Presentation

The 8-month-old girl, the firstborn of unrelated healthy parents, was reported to have normal growth and development. She was taken to the emergency room due to the mother’s at-home observation of a state of hyporeactivity and hypotonia. The mother also reported attempted breastfeeding with difficulty in sucking due to incoordination. The little girl attempted to stand but fell backwards, hitting her occiput and immediately crying, with no loss of consciousness or vomiting. No fever or any other significant symptoms were reported in the previous days.

At a physical examination, the child was drowsy and rousable for brief moments with persistent crying, hypotonia, and bilateral miosis. Upon palpation of the skull, no swellings or bony steps were observed. The child had pinkish, pale skin; a pernicious, diffusely harsh vesicular murmur in the chest with good air penetration; and strong and rhythmic cardiac activity. Her abdomen was treatable. She was afebrile, with an SpO_2_ of 99% in room air, an ABP of 99/51 mmHg, and a heart rate of 113/min.

The laboratory workup was not informative. The PCR on the pharyngeal swab used for the detection of the SARS-CoV-2 genome was negative. The electrocardiogram and electroencephalogram were normal. The computed tomography scan of the skull and brain was normal. The toxicology study of blood and urine was completely negative. The Flumazenil administration was not effective in reducing the child’s state of drowsiness. Starting from hour +4, the child progressively improved, and by hour +6, she recovered to a normal state of consciousness.

In an attempt to identify a possible toxic agent, the parents were asked to check the home environment within the child’s reach. Later on, the mother reported that the vials of brimonidine-based eye drops used by the grandfather were found empty on the sofa near where the little girl had been playing.

## 3. Biochemical Study

Following the suspicion of brimonidine consumption through eye drops, ultra-performance liquid chromatography–tandem mass spectrometry (UPLC-MS/MS) was applied for the determination of brimonidine’s presence in urine and plasma samples, respectively [23,24], collected about 10 h after exposure. The UPLC-MS/MS consisted of an ACQUITY™ UPLC™ I-Class system (comprising a binary solvent manager (BSM) and a sample manager with a flow-through needle (SM-FTN)) coupled to a Xevo^®^ TQ-S micro mass spectrometer (Waters Corporation, Milford, MA, USA).

A total of 100 μL of plasma/urine study samples, calibrators, and QCs were added to 300 μL of IS working solution (acetonitrile containing IS 2 ng/mL for plasma sample processing and 100 ng/mL for urine sample processing). After centrifugation, supernatants were transferred into other tubes and then evaporated until dry. The residues were reconstituted with 100 μL of mobile phase and finally transferred into vials for UPLC-MS/MS analysis. Altogether, 10 μL of the sample was injected into the ion source. More details on the calibrator and quality control materials, sample preparation, and UPLC-MS/MS analysis are fully reported in the Appendix A.

The UPLC-MS/MS system operated in positive electrospray ionization (ESI+). The run time was 6 min, injection-to-injection, using an ACQUITY UPLC^®^ BEH C18 Vanguard pre-column and an ACQUITY UPLC^®^ BEH C18 2.1 mm × 150 mm, 1.7 μm column (T = 40 °C). The mobile phase was composed of water (A) with 0.1% formic acid and acetonitrile (B) with 0.1% formic acid (B). The flow rate was 0.5 mL/min. The LC gradient is elucidated in Appendix A. Moreover, parameters referring to MRM functions for the detection of brimonidine are reported in Appendix A. The data were processed using TargetLynx™ XS software V 4.2 (Waters Corporation, Milford, MA, USA). Chromatograms related to blank and spiked samples of urine and plasma sample types, respectively, are shown in Appendix A.

The UPLC/MS/MS analysis performed on the urine sample revealed 8.40 ng/mL brimonidine concentration levels, supporting the initial clinical suspicion. The chromatograph in Figure 2 clearly highlights the presence of brimonidine in the patient’s urine sample.

The clinical suspicion was further supported by the analysis of the patient’s plasma sample with brimonidine concentration levels of 0.79 ng/mL, as shown in the chromatograph reported in Figure 3.

## 4. Discussion

Brimonidine is a potentially lethal medication for young infants, even as an ophthalmic drop. The mechanisms of neurotoxicity associated with brimonidine are not fully elucidated, but some hypotheses exist. Brimonidine is a selective agonist of alpha-2 adrenergic receptors. While this action primarily leads to reduced intraocular pressure by decreasing aqueous humor production and increasing uveoscleral outflow, the overstimulation of alpha-2 adrenergic receptors in the central nervous system could lead to adverse effects. This might include sedation, depression of the respiratory drive, and, potentially, neurotoxicity [11,12,13]. Another possible mechanism is that the activation of alpha-2 adrenergic receptors in the central nervous system can lead to sedation and decreased alertness [25]. In high doses, brimonidine may cause CNS depression, potentially leading to neurotoxicity. Some studies suggest that brimonidine may induce oxidative stress in neuronal cells. Oxidative stress occurs when there is an imbalance between the production of reactive oxygen species (ROS) and the ability of the body to detoxify them or repair the resulting damage. This oxidative stress can lead to the damage of cellular components, including proteins, lipids, and DNA, ultimately leading to neurotoxicity [26]. Glutamate is the primary excitatory neurotransmitter in the central nervous system. Excessive glutamate release or impaired glutamate clearance can lead to neuronal damage or death. Some research suggests that brimonidine may modulate glutamate neurotransmission, potentially contributing to neurotoxic effects [27]. Another possible mechanism of toxicity is connected with calcium dysregulation. Calcium ions play crucial roles in neuronal signaling and cell survival. The disruption of calcium homeostasis can lead to neuronal dysfunction and death. Some studies suggest that brimonidine may affect calcium channels or intracellular calcium levels, potentially contributing to neurotoxicity. Finally, brimonidine may also modulate inflammatory responses in the central nervous system. Chronic inflammation is associated with various neurological disorders, and the modulation of inflammatory pathways by brimonidine could potentially contribute to neurotoxicity [28].

Studies on different animal models have shown that topical solutions cause greater systemic absorption than topical gels [29]. It has been reported that tear volumes increase with age, being 0.5 μL for newborns, 2.5 μL for older infants, and 6 μL for adults [26]; thus, topically applied drugs are much more concentrated in young infants. Based on these data, a newborn needs only one-half and a 3-year-old child only needs two-thirds of the adult dosage to reach an equivalent ocular concentration [19]. The eye membranes in infants are thinner than in adults, allowing for faster drug absorption and corneal permeation. Moreover, the drug solution can reach the nasal cavity, where the compound can be absorbed through the nasal mucosa and reach systemic circulation, bypassing the liver and thus avoiding first-pass metabolism [30].

In 2021, Ghaffari et al. reported a series of six cases found in the literature on the systemic side effects of the accidental oral or nasal ingestion of brimonidine eye drops in children, reported up to the end of 2019 [31], together with five additional new cases from personal observation. Their ages ranged between 9 days and 4 years.

Miosis may be a helpful sign when investigating brimonidine intoxication but should not be considered a must for confirming the diagnosis.

The duration of hypotonia may be variable. Ghaffari et al. report that in all of their cases, hypotonia lasted about twice the time needed for recovery to a normal level of consciousness [31].

In the present case, the child was restored by hour +6 to a normal level of consciousness and muscular tone. The duration and type of complication may (also) likely reflect the dose of brimonidine consumed [17,32]. Ghaffari et al. acknowledge that a lack of body fluid brimonidine concentration was a limitation in their study [31]. In our case, the possibility of revealing the presence of brimonidine and accurately measuring its concentration levels through a specific and sensitive approach based on the UPLC-MS/MS analysis of blood and urine samples, respectively, allowed us to promptly confirm the diagnosis of accidental brimonidine intoxication.

At the most recent follow-up visit, 11 months after the reported event, the child was doing fine, her parents reported a series of typical age-related behaviors, and we found no evidence of a residual defect or alteration upon careful physical examination.

Following the ocular administration of one drop of brimonidine (0.2%) in adult patients in each eye twice a day, a maximal plasma drug concentration is reported to be 0.0585 ng/mL [19,33]. The drug’s absorption is known to occur within 1–4 h with an elimination half-life of about 3 h. Systemic brimonidine undergoes an extensive liver metabolism, and the drug, with its metabolites, is eliminated mainly by urinary excretion [8]. Approximately 74% of an orally administered radioactive dose of brimonidine is recovered in the urine 120 h after the treatment.

The brimonidine concentration in the plasma sample collected from our patient about ten hours after its accidental ingestion was 0.790 ng/mL. Thus, in our 8-month-old girl, plasma concentrations were about 13-fold higher than the maximum mean plasma concentration observed in adults. Of course, it has to be remarked that this is oral versus topical consumption.

This level was similar to that found in an infant with Peters anomaly being treated with ophthalmic drops containing 0.2% brimonidine. The young patient was brought to the emergency department, proving lethargic, hypotonic, hypothermic, and unresponsive to stimulation. These episodes were repeated five times during the hospitalization period. In this case, brimonidine concentrations were reported to be 1.459 ng/mL and 0.700 ng/mL in the plasma 0–3 and 6 h after instillation, respectively [19]. These elevated plasma levels, along with the resolution of symptoms following brimonidine withdrawal, indicated the drug as the possible cause of the reported intermittent coma episodes [19].

Accidental and non-intentional self-ingestion still presents as a major mode of childhood home poisoning. While ingestion is evident and well documented in some cases, in other cases, it may be supposed by indirect evidence or based on clinical manifestations and a lack of other explanations. In the present case, we report one example of clinical and laboratory cooperation aimed at documenting the origin of a frightening clinical scenario caused by a largely preventable inadvertent social behavior.

## 5. Conclusions

To our knowledge, this is the first report to determine brimonidine levels in urine and plasma using UPLC-MS/MS.

Insufficient knowledge on the part of family members about the potential hazards of an apparently innocuous, topical medication such as eye drops may put children at a greater risk of poisoning. Among the reasons for the higher vulnerability observed among young infants, an immature blood–brain barrier and deficiency of specific cytochrome P450 enzymes can be included [34,35]. Necessary warnings should be given to parents with greater care when prescribing this medication.

## Figures and Tables

**Figure 1 children-11-00317-f001:**
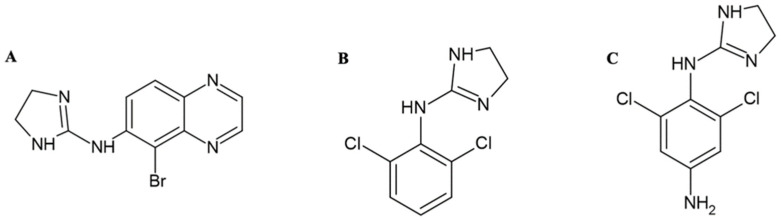
Chemical structure of (**A**) brimonidine, IUPAC name: 5-bromo-N-(4,5-dihydro-1H-imidazol-2-yl)quinoxalin-6-amine; (**B**) clonidine, IUPAC name: N-(2,6-diclorofenil)-4,5-diidro-1H-imidazol-2-amina; (**C**) apraclonidine, IUPAC name: 2,6-dichloro-1-N-(4,5-dihydro-1H-imidazol-2-yl)benzene-1,4-diamine. Created using ACD/ChemSketch Freeware version 2023.1.2, accessed date 23 January 2024 (Advanced Chemistry Development, Inc., Toronto, ON, Canada).

**Figure 2 children-11-00317-f002:**
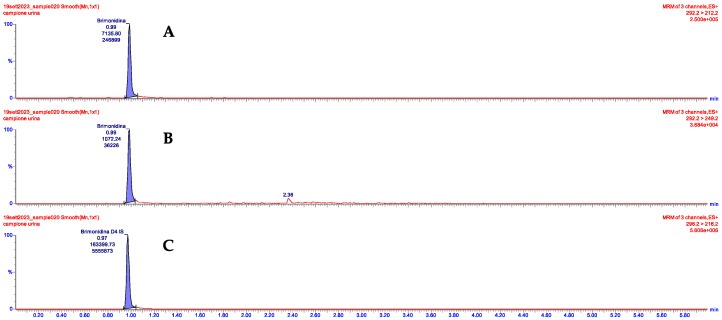
Chromatographic separation of (**A**) brimonidine (quantifier); (**B**) brimonidine (qualifier); and (**C**) internal standard brimonidine-d4 in urine sample of the study child.

**Figure 3 children-11-00317-f003:**
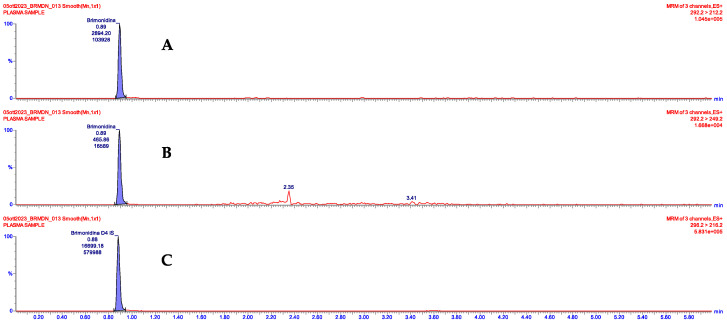
Chromatographic separation of (**A**) brimonidine (quantifier); (**B**) brimonidine (qualifier); and (**C**) internal standard brimonidine-d4 in plasma sample of the study child.

## Data Availability

The data presented in this study are available in article and Appendix A.

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
