# Peer review of "Brimonidine Eye Drops within the Reach of Children: A Possible Foe"

_children, 2024, doi:10.3390/children11030317_

Round 1

Reviewer 1 Report

Comments and Suggestions for Authors

Rarely do I ever read a drafted manuscript that is ready for publication in its un-edited post-referee form. This submission, in my opinion, is that. The literature is replete with citations regarding the risk of brimonodine in a pediatric population in topical therapy. The inherent risk of ingesting the drug is worthwhile of immediate publication. There is risk to the public and a need for parents and care providers to acknowledge.

Author Response

We thank the reviewer for his/her favorable evaluation. 

Reviewer 2 Report

Comments and Suggestions for Authors

The study “Brimonidine eye drops at the children reach: a possible foe” presents a case of transient encephalopathy in a toddler due to accidental brimonidine toxicity, a medication used for open-angle glaucoma. Despite initial negative tests, subsequent investigation using UPLC-MS/MS revealed brimonidine in urine and plasma. The child's condition improved within hours. This highlights the risk of insufficient awareness among families regarding the hazards of seemingly harmless topical medications, urging the need for enhanced communication and warnings when prescribing such drugs to parents.

The case study provides valuable insights into the potential neurological side effects of brimonidine, shedding light on a scenario that may be underreported or underestimated. The use of UPLC-MS/MS for determining brimonidine levels in urine and plasma adds a novel dimension to the study.

Furthermore, the emphasis on the need for increased awareness among families regarding the risks associated with seemingly innocuous topical medications, like eye drops, is crucial for preventing accidental poisoning in children.

It is recommended to enlarge the text in the figures, since it is impossible to read and you have to zoom in a lot.

I recommend making some minor changes:

In the abstract: define all abbreviations (ER, CT hour+4, hour +6))

Same for the introduction, case presentation (AMP, ECG, EEG, CT).

Discussion: lines 152 and 164 “hrs” replace by hours.

Author Response

Reviewer 2

  • It is recommended to enlarge the text in the figures, since it is impossible to read and you have to zoom in a lot
    • Re: We would like to specify that chromatographic separations as shown in Figure 2 and 3 are directly generated by the software of the UPLC-MS/MS after the analysis. Anyway, to make it more readable, we inserted both in Figure 2 (urine sample) and Figure 3 (plasma sample) the letters A, B, and C to indicate brimonidine (quantifier), brimonidine (qualifier) and internal standard brimonidine-d4, respectively. Figure legends were also updated.

  • In the abstract: define all abbreviations (ER, CT hour+4, hour +6))
    • Re: This was done.
  • Same for the introduction, case presentation (AMP, ECG, EEG, CT).
    • Re: This was done.
  • Discussion: lines 152 and 164 “hrs” replace by hours.
    • Re: This was done.

Reviewer 3 Report

Comments and Suggestions for Authors

In the present document, the authors aim to report a case in which an 8-month-old infant presented to the emergency healthcare department due to intoxication with Brimonidine drops. From my perspective, the report is intriguing, well-written, and well-structured. Please find enclosed below comments that I hope will enhance the reliability of the manuscript.

Lines 38-39 – To provide more context to the presented report, it could be beneficial to add one or two sentences regarding the prevalence or incidence of Open-Angle Glaucoma (OAG) in both adults and children.

Line 43 – Define the acronym "AMP."

Line 54 – The brand and manufacturer of ACD/ChemSketch should be reported.

Line 105 - The brand and manufacturer should be reported.

Lines 170-178 – To lend more relevance to the present report, it could be useful to add one or two sentences with clinical implications or contribute to the ongoing scientific debate related to the presented case report within the scientific community.

Comments on the Quality of English Language

No Comments

Author Response

  • Lines 38-39 – To provide more context to the presented report, it could be beneficial to add one or two sentences regarding the prevalence or incidence of Open-Angle Glaucoma (OAG) in both adults and children.
    • Re: this was done
  • Line 43 – Define the acronym "AMP."
    • Re: this was done
  • Line 54 – The brand and manufacturer of ACD/ChemSketch should be reported.
    • Re: this was done
  • Line 105 - The brand and manufacturer should be reported.
    • Re: this was done
  • Lines 170-178 – To lend more relevance to the present report, it could be useful to add one or two sentences with clinical implications or contribute to the ongoing scientific debate related to the presented case report within the scientific community.
    • Re: This was done

Reviewer 4 Report

Comments and Suggestions for Authors

In this case report, a severe central nervous system adverse reaction to an alpha2 agonist glaucoma droplet used completely off-label in an 8-month-old child is reported. Unfortunately, this may have pharmaceutical company and medicolegal implications.

How could this happen? Whichever drug label is included (Novartis, Allergan/Abbvey: Simbrinza, Alphagan, Combigan), the use of brimonidine is contraindicated under two years of age.

If the Editorial Board decides to publish this case study, my recommendations are as follows:

In the Introduction, all known side effects of the drug should be described, as well as the companies' recommendations for its use in children.

In the Conclusions, it should be emphasized that the use of brimonidine is contraindicated in all company brochures for children under 2 years of age and that caution should be practiced between the ages of 2 and 17 years.

Author Response

Reviewer 4

  • In this case report, a severe central nervous system adverse reaction to an alpha2 agonist glaucoma droplet used completely off-label in an 8-month-old child is reported. Unfortunately, this may have pharmaceutical company and medicolegal implications.
  • How could this happen? Whichever drug label is included (Novartis, Allergan/Abbvey: Simbrinza, Alphagan, Combigan), the use of brimonidine is contraindicated under two years of age.
  • If the Editorial Board decides to publish this case study, my recommendations are as follows:
  •  
  • In the Introduction, all known side effects of the drug should be described, as well as the companies' recommendations for its use in children.
  • In the Conclusions, it should be emphasized that the use of brimonidine is contraindicated in all company brochures for children under 2 years of age and that caution should be practiced between the ages of 2 and 17 years.

Re.: We are afraid that the worries of the reviewer may derive from a misunderstanding. In the MS we clearly state that the assumption of brimonidine by the child was completely undesired. Indeed, in the last paragraph of the case report, we state as follows: “In an attempt to identify a possible toxic agent, the parents were asked to check the home environment at the child's reach. Later on, the mum reported the vials of brimonidine-based eye drops used by the grandfather, found empty on the sofa near where the little girl had been playing.” Thus, the worries on incorrect prescription of brimonidine to the child apper not to be supported in this case.

Anyway, the following statement was added at the end of the discussion:

Accidental and non-intentional self-ingestion still presents as a major mode of childhood home poisoning. While in some cases ingestion is evident and well documented, in other cases it may be supposed by indirect evidence or based on clinical manifestations and lack of other explanation. In the present case, we report one example of clinical and laboratory cooperation aimed at documenting the origin of a scaring clinical picture as caused by a largely preventable inaccurate social behavior.

Reviewer 5 Report

Comments and Suggestions for Authors

February 17 2024

Re: Brimonidine eye drops at the children reach: a possible foe

Manuscript ID: children-2871963

Dear authors,

This case study examines the severe side effects experienced by a pediatric patient after accidentally ingesting brimonidine eye drops, a medication typically used to reduce intraocular pressure in glaucoma patients. This incident highlights the risks posed by the accessibility of such medications to children, emphasizing the need for preventive measures to safeguard them. Although the case is clearly presented, it requires several revisions, especially in the discussion section, which should be expanded to at least twice its current. Incorporating these changes will strengthen the article's contribution to understanding the implications of brimonidine use in pediatric patients and the importance of stringent safety measures.

References

Please seek more robust references than those currently cited as numbers 11 and 16:

Montague A., Bangh S., Cole J., eds. "A Toddler’s Toxic Taste: Bradycardia After Brushing with Brimonidine." American College of Medical Toxicology (ACMT) Annual Scientific Meeting 2017.

PdfDownloadServlet (agenziafarmaco.gov.it) accessed on December 15, 2023.

Figures

Figure 1 is blurry.

Discussion on Mechanisms

The focus on the neurological side effects of brimonidine requires a deeper exploration of the pharmacological mechanisms that contribute to these side effects and hazards.

Discussion Volume

The discussion should be significantly expanded to provide a thorough analysis and interpretation of the findings, which will deepen the understanding of the case's implications. Additionally, the methodology of the toxicology study should be detailed to enhance the discussion. It is suggested to conclude the discussion with specific preventive measures or guidelines.

Long-term Follow-Up

The article lacks information on the long-term follow-up of the patient's condition post-exposure, including aftercare and follow-up details.

Comments on the Quality of English Language

Minor to Moderate editing of English language required

Author Response

Reviewer 5

Dear authors,

This case study examines the severe side effects experienced by a pediatric patient after accidentally ingesting brimonidine eye drops, a medication typically used to reduce intraocular pressure in glaucoma patients. This incident highlights the risks posed by the accessibility of such medications to children, emphasizing the need for preventive measures to safeguard them. Although the case is clearly presented, it requires several revisions, especially in the discussion section, which should be expanded to at least twice its current. Incorporating these changes will strengthen the article's contribution to understanding the implications of brimonidine use in pediatric patients and the importance of stringent safety measures.

  1. Please seek more robust references than those currently cited as numbers 11 and 16:
  • : Ref. 11 PdfDownloadServlet (agenziafarmaco.gov.it) accessed on December 15, 2023. was replaced by the following: Boccaccini, A., Cavaterra, D., Carnevale, C., Tanga, L., Marini, S., Bocedi, A., Lacal, P. M., Manni, G., Graziani, G., Sbardella, D., & Tundo, G. R. (2023). Novel frontiers in neuroprotective therapies in glaucoma: Molecular and clinical aspects. Molecular aspects of medicine, 94, 101225. https://doi.org/10.1016/j.mam.2023.101225.
  • : Ref. 16 Montague A., Bangh S., Cole J., eds. "A Toddler’s Toxic Taste: Bradycardia After Brushing with Brimonidine." American College of Medical Toxicology (ACMT) Annual Scientific Meeting 2017. This was omitted.

  1. Figure 1 is blurry.
    1. : As requested, we inserted in the revised version of the manuscript a Figure 1 with a higher resolution.

  1. Discussion on Mechanisms. The focus on the neurological side effects of brimonidine requires a deeper exploration of the pharmacological mechanisms that contribute to these side effects and hazards.
    1. : As suggested, possible mechanisms of neurotoxicity of brimonidine have been widely mentioned in the discussion.

  1. Discussion Volume. The discussion should be significantly expanded to provide a thorough analysis and interpretation of the findings, which will deepen the understanding of the case's implications. Additionally, the methodology of the toxicology study should be detailed to enhance the discussion. It is suggested to conclude the discussion with specific preventive measures or guidelines.

  1. : As suggested by the reviewer, the volumen of the discusion has been markedly expanded by discussion of the mechanisms of neurotoxicity and also by a comment on the methodology used in the study, which was also added.

  1. Long-term Follow-Up. The article lacks information on the long-term follow-up of the patient's condition post-exposure, including aftercare and follow-up details.
    1. : The following information was introduced: “At the most recent follow-up visit, 11 months after the reported event, the child is doing fine, her parents report a fully age-related behavior and we found no evidence of residual defect or alteration at careful physical examination.”

Reviewer 6 Report

Comments and Suggestions for Authors

The authors present a well-written case report of brimonidine toxicity in an 8-month-old, with quantification of urine and plasma levels.

Could the authors add a bit more context to the discussion?  In the discussion, the authors mention that the plasma concentration in the toddler was 13 times higher than that seen win adults.  I would suggest clarifying that this is oral vs topical.  Also, the plasma levels were similar to that seen with topical application in the infant with Peters anomaly.  But this was plasma levels seen at approximately 10 hours from oral exposure in this case versus 6 hours from topical exposure in the infant with Peters anomaly.  The authors mention that the half-life of brimonidine is about 3 hours - is that systemic half-life?  Would it then be fair to conjecture that the plasma levels at 6 hours (one half life prior) may have been twice as high, and thus also twice as high as that seem with topical administration in the case reported by Berlin et al?

Comments on the Quality of English Language

This is a well-written manuscript, engendering very few comments.

I would suggest editing the title to: "Brimonidine eye drops within the reach of children: a possible foe"

I would remain consistent with the decimal notation, e.g. 0.79ng/mL and 0.2% versus 0,790 ng/mL (line 157) and 0,2% (line 149).  Though not a must, you may also want to consider converting the plasma levels reported by Berlin et al from pg/ML to ng/ML, for the sake of consistency.

Line 154-5: consider changing "brimonidine is recovered in urine after 120 hours from treatment" to  "brimonidine is recovered in urine 120 hours after treatment"

Author Response

  • Could the authors add a bit more context to the discussion?  In the discussion, the authors mention that the plasma concentration in the toddler was 13 times higher than that seen win adults.  I would suggest clarifying that this is oral vs topical. 
    • : this was done

  • Also, the plasma levels were similar to that seen with topical application in the infant with Peters anomaly.  But this was plasma levels seen at approximately 10 hours from oral exposure in this case versus 6 hours from topical exposure in the infant with Peters anomaly.  The authors mention that the half-life of brimonidine is about 3 hours - is that systemic half-life?  Would it then be fair to conjecture that the plasma levels at 6 hours (one half life prior) may have been twice as high, and thus also twice as high as that seem with topical administration in the case reported by Berlin et al?
    • : Yes, 3 hours refers to the brimonidine systemic half-life. We made the same“conjecture” too, but we decided not to insert it in the MS. It would have been necessary to carry out measurements at different times in order to provide precise data on which to propose a comment, also considering the peculiarity of drug clearance (either renal or hepatic) in neonates (1 month in the case of Peters anomaly) and young infants (8 months in our case) where the developmental differences produced by the impact of ontogeny on renal functions and drug metabolizing enzymes activity deeply modify drug disposition and action.
  • I would suggest editing the title to: "Brimonidine eye drops within the reach of children: a possible foe"
    • : this was done

  • I would remain consistent with the decimal notation, e.g. 0.79ng/mL and 0.2% versus 0,790 ng/mL (line 157) and 0,2% (line 149). 
    • : this was done

  • Though not a must, you may also want to consider converting the plasma levels reported by Berlin et al from pg/ML to ng/ML, for the sake of consistency.

  • : this was done

  • Line 154-5: consider changing "brimonidine is recovered in urine after 120 hours from treatment" to  "brimonidine is recovered in urine 120 hours after treatment"

  • : this was done

Round 2

Reviewer 4 Report

Comments and Suggestions for Authors

The manuscript contains very important information regarding the dangers  of using brimonidine eyedrops in children

Author Response

Reviewer 4:

The manuscript contains very important information regarding the dangers  of using brimonidine eyedrops in children.

Re.: thank you for your favorable comment. We remark, once more, that assumption of brimonidine in the present case was fully inadvertent inasmuch the eyedrop medication (used by his grandafther) happened to be within the reach of the child.  

Reviewer 5 Report

Comments and Suggestions for Authors

The authors did not respond to comments

Comments on the Quality of English Language

Minor editing of English language required

Author Response

We apologize for any misunderstanding. The remarks to the "round 1" evaluation have now been uploaded above. 

Round 3

Reviewer 5 Report

Comments and Suggestions for Authors

The authors have submitted a revised version of the manuscript, addressing all comments . The article is now in compliance with the required standards.

Comments on the Quality of English Language

Minor editing